# A Hybrid Deep Learning Model as the Digital Twin of Ultra-Precision Diamond Cutting for In-Process Prediction of Cutting-Tool Wear

**Lei Wu, Kaijie Sha, Ye Tao, Bingfeng Ju and Yuanliu Chen \***

The State Key Laboratory of Fluid Power and Mechatronic Systems, Zhejiang University, Hangzhou 310027, China
\* Correspondence: yuanliuchen@zju.edu.cn

**Abstract:** Diamond cutting-tool wear has a direct impact on the processing accuracy of the machined surface in ultra-precision diamond cutting. It is difficult to monitor the tool's condition because of the slight wear amount. This paper proposed a hybrid deep learning model for tool wear state prediction in ultra-precision diamond cutting. The cutting force was accurately estimated and the wear state of the diamond tool was predicted by using the hybrid deep learning model with the motion displacement, velocity, and other signals in the machining process. By carrying out machining experiments, this method can classify diamond-tool wear condition with an accuracy of more than 85%. Meanwhile, the effectiveness of the proposed method was verified by comparing it with a variety of machine learning models.

**Keywords:** ultra-precision diamond cutting; cutting-tool wear; in-process prediction; digital twin; deep learning

## 1. Introduction

Ultra-precision diamond cutting has been widely used because of its advantages of high precision and fast speed. In the process of ultra-precision cutting, diamond tool wear (DTW) is an unavoidable physical phenomenon and influences high-precision surface quality [1,2]. Even tiny DTW can have a direct influence on the machining accuracy and surface quality of the machined surface. Especially in the processing of hard brittle materials or special large parts, DTW will become the bottleneck by limiting the qualified rate and performance of parts [3]. To avoid product failure and cost loss caused by DTW, it is important to establish a set of tool condition monitoring (TCM) systems to monitor the tool condition. TCM can maximize the life of diamond tools and minimize the accuracy influence caused by DTW.

There are two types of approaches for predicting tool wear, namely, model-based approaches and data-driven approaches. In contrast to model-based approaches, data-driven approaches model the data utilizing a learning process, avoiding any assumptions of their underlying distribution [4], as well as the influence of the hypothesis on the accuracy of the model [5]. Many data-driven approaches for tool wear prediction are developed based on machine learning such as artificial neural networks (ANN), support vector machines (SVM), and decision trees (DT) [6]. Conventional machine learning is limited in its ability to process raw data, which has a negative influence on their generalization to normal machining conditions [7]. In tool condition monitoring under complex and changeable conditions, the deep learning model can overcome the shortcomings of traditional machine-learning-based monitoring approaches and obtain higher monitoring accuracy.

Based on the background described above, this paper presented a digital twin system [8] based on a hybrid deep learning (HDL) model to predict the wear state of diamond tools in the ultra-precision machining process. The digital twin system can monitor the wear condition of the diamond tool in real time and ensure the machining accuracy of the

products in the processing. The high-dimensional feature extraction of the HDL model has a stronger generalization ability for the complex machining environment. Therefore, the digital twin system will have higher monitoring accuracy. The effectiveness of the proposed HDL model was verified by comparing it with a variety of machine learning models.

The rest of this paper is organized as follows. Section 2 reviews related work. Sections 3 and 4 present the proposed approach and its key enablers, respectively. Section 5 includes the experiments and discussions on the disadvantages of the proposed approach. Section 6 draws the conclusion.

## 2. Related Work

The method of tool wear estimation can be divided into direct detection and indirect detection. Direct detection (e.g., optical measurement [9–11], scanning electron microscope (SEM) measurement [12], and atomic force microscope (AFM) measurement [13]) measures the tool surface state, the wear amount of the tool surface, and other parameters directly, to confirm the wear state of the diamond tool. However, its practical application is restrained by the cutting fluid, chip, lighting, and other conditions in the processing. Therefore, it can be applied to field monitoring, but online monitoring or in situ monitoring is difficult [3]. The optical measurement approach is more suitable for industrial applications, although the resolution range is limited to several hundred nanometers by optical diffraction [3].

On the other hand, the indirect detection approach judges the tool wear condition indirectly through the cutting force, surface roughness, vibration, acoustic emission, and other signals in the processing. This approach has lower accuracy, but the signals are easy to monitor online. Indirect detection can be divided into four main steps, namely, signal acquisition, signal processing, feature extraction, and state recognition. For example, in the aspect of signal acquisition, Hossein et al. [14,15] detected acoustic emission signals and represented DTW with signals and cutting distance. Yamaguchi et al. [16] used broadband acoustic emission sensors to measure the acoustic emission signals during the processing of nickel–phosphorus alloy by diamond tools. Cutting force signals contain characteristic information about tool wear and are often used to monitor DTW in ultra-precision diamond cutting [17]. Yamaguchi et al. [16] measured the thrust variation during the processing of nickel–phosphorus alloy by force sensors, and further deduced DTW by identifying the $1/f\beta$ noise in the power spectrum which varies with the degree of tool wear. Yan et al. [17] confirmed the possibility of using a cutting force signal to monitor tool wear. Choi et al. [18] analyzed the tool wear state by using cutting force characteristics such as phase-frequency characteristics and power amplitude-frequency characteristics of cutting force signals in the cutting process. Zong et al. [19] proved through experiments that DTW would lead to an increase in cutting force. In signal processing and feature extraction, the wavelet transform technique can effectively detect transient features such as singularity or discontinuity, which greatly reduces the signal processing time [3]. In the state recognition, Scheffer et al. [20] collected cutting force signals and vibration signals in ultra-precision machining, extracted wear-related signals by correlation analysis, and identified the wear state by self-organizing mapping (SOM). Ko et al. [21] identified the state of the DTW by the cutting force signals and a fuzzy pattern recognition network.

Due to the increase in machining data volume, deep learning algorithms have attracted more and more attention. Arellano et al. [4] performed time series images on the forces in three dimensions by time series imaging method and fed the three-channel images into a convolutional neural network for tool wear classification. Another way to obtain image-based datasets from time series is by imaging signals by time-frequency analysis techniques such as wavelet transform. Then, the resulting energy spectrum pictures are fed into the convolutional neural network. Xu et al. [22] proposed a side-edge wear depth model based on feature fusion. They converted the signals of multiple sensors into multi-channel images, and then fed the multi-channel images into different convolutional neural networks to extract features in parallel from multi-source data and make tool-wear predictions.

There are still some problems with DTW monitoring to be solved. The online monitoring of the DTW state requires feature extraction of machining signals, but the signals tend to be slight while DTW in ultra-precision diamond cutting is usually a small amount, which makes feature extraction difficult. Establishing a digital twin of the DTW is an important approach to realizing prediction-based fast manufacturing optimization [8,23,24]. Due to the demands of ultra-precision machining, the machining morphology and tool trajectory tend to be complex, which leads to the complexity and diversity of machining signals. This further leads to an increase in the difficulty of identifying tool wear and establishing the digital twins through machining signals.

## 3. The Hybrid Deep-Learning-Model-Based Diamond-Tool Wear Prediction

This paper developed a demonstrative ultra-precision diamond cutting machine tool, in which the fast tool servo (FTS) machining system adopts the extra fast servo tool installed on the Z-axis of the machine tool to realize the fast Z-direction feed and completes the surface topography processing. The fast tool servo device adopted in this paper is designed with a voice coil motor as the driver, grating ruler as the position feedback sensor, flexible hinge as the transmission and stop mechanism of large stroke, and high-precision electromagnetic drive fast tool servo device. Figure 1 is a schematic diagram of its overall mechanism.

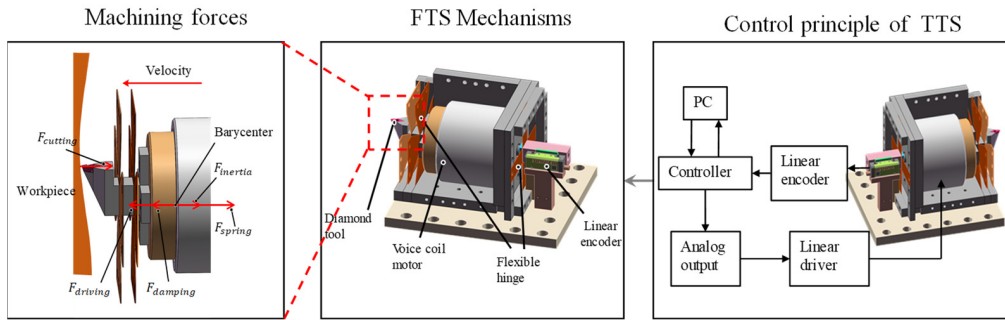

**Figure 1.** Schematic diagram of the overall mechanism of FTS [25].

In the digital twin system, state recognition is the core part of the system, which directly determines the accuracy of the system. To recognize the wear state of diamond tools, feature extraction of machining signals is needed. However, due to the slight change of signals caused by tiny DTW, it is very difficult to manually complete the feature extraction process. The adaptive feature extraction of the deep learning method can also reasonably solve the difficulty of feature extraction caused by slight signal changes. This paper proposed an HDL (i.e., hybrid deep learning) model to predict the wear state of diamond tools. Furthermore, the TCM system is established to realize the online monitoring of diamond tools. The flow diagram of HDL-based diamond-tool prediction is shown in Figure 2.

In the process of tool monitoring, the cutting force contains a lot of information and characteristics about tool wear, so it is applied in the monitoring of the tool state. However, the force sensor (including the piezoelectric approach and strain approach) is not easy to install because of the high cost of measurement, which is the reason for other ways to predict the tool state in this paper. A self-sensing cutting force estimation method is established based on machine learning. According to the machining signal and control signal of the FTS device in the ultra-precision machining process, the cutting force signal is estimated by using a multi-layer fully connected neural network trained by the special machining process signal. This method does not need to rely on a force sensor and can achieve an accurate estimation of cutting force.

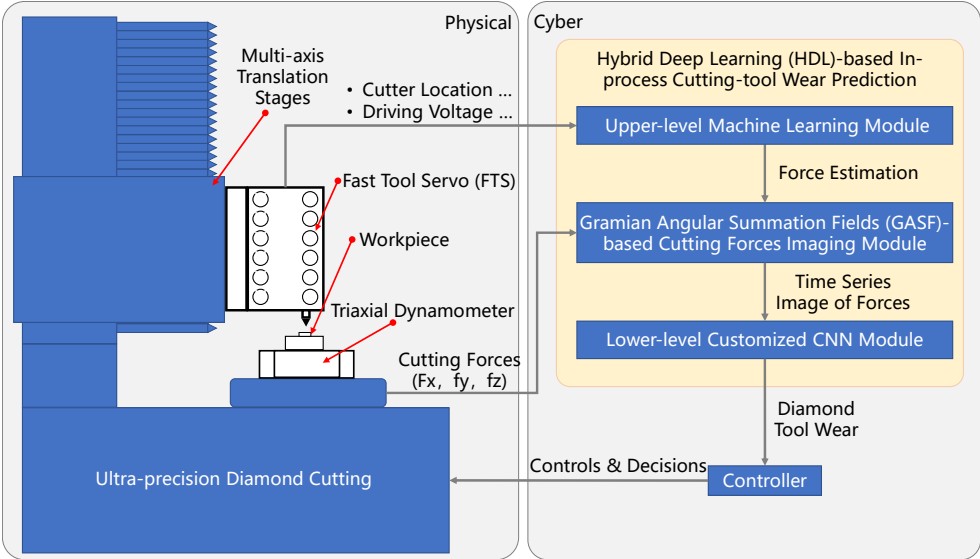

**Figure 2.** Flow diagram of HDL-based diamond-tool prediction.

Combined with the cutting force estimation method, machining signals of the FTS device are used to estimate and reconstruct cutting force. Since the convolutional neural network is sensitive to the spatial characteristics of the signal, a one-dimensional signal of estimated cutting force is converted into a two-dimensional signal by the time series imaging method. The transformed two-dimensional image data are input into the HDL model for training, and the trained state recognition effect is verified. To train the HDL model for diamond-tool wear state prediction, a large number of high-quality data samples containing different wear states of diamond tools need to be obtained through a large number of experiments.

## 4. Key Enablers in the HDL Model

### 4.1. The Upper-Level Machine Learning Module for Cutting Force Estimation

When the FTS device is used for diamond cutting, there will be a variety of machining forces including Lorentz force ($F_{driving}$), spring force ($F_{spring}$), inertia force ($F_{inertia}$), damping force ($F_{damping}$), and cutting force ($F_{cutting}$). Lorentz force is the force generated by the voice coil motor that is used to drive the device to cut, so it can also be called the driving force. Spring forces are produced by the deformation of elastic hinges. The acceleration of the moving parts results in the presence of inertial forces. Damping forces are caused by the air and the inevitable friction within the device. Cutting force is the resistance to deformation generated during the process of material deformation and removal. Their relationship is complex. Due to the existence of the above forces, it is difficult to accurately obtain the cutting force of the interaction between the diamond tool and the workpiece surface along the axial direction of the system. The driving force, spring force, damping force, and inertia force of the device can be expressed as follows:

$$F_{driving} = B{\cdot}I{\cdot}L{\cdot}n = B{\cdot}L{\cdot}n{\cdot}p{\cdot}V, \tag{1}$$

$$F_{spring} = k{\cdot}x, \tag{2}$$

$$F_{damping} = c{\cdot}\dot{x}, \tag{3}$$

$$F_{inertia} = m{\cdot}\ddot{x}, \tag{4}$$

where $B$ represents the magnetic field intensity of the voice coil motor; $I$ represents the current passing through the voice coil motor; $L$ represents the length of one turn of the coil; $n$ represents the number of turns of the voice coil motor; $V$ represents the driving voltage

signal used for the voice coil motor; $p$ represents the proportional coefficient between the driving voltage and the output current; $x$ represents the position of the tool; $k$ represents the spring coefficient of the elastic hinge; $c$ represents the sum of damping coefficients in the air and device; and $m$ represents the mass of the moving parts.

Utilizing the relationship between each machining force in the force equation, the cutting force can be calculated according to the machining signal (e.g., tool displacement signal) and control signal (e.g., drive voltage signal) in the machining process of the device, to achieve the estimation of cutting force. However, due to the uneven distribution of the magnetic field in the range of travel of the voice coil motor, the intensity of the magnetic field is related to the position of the tool, so the driving current and the driving force are not linear. The corresponding relationship between the two is a nonlinear relationship coupled with the displacement of the tool, which is shown in the following equation.

$$F_{driving} = f(x) \cdot I = f(x) \cdot pV. \tag{5}$$

This relationship hinders accurate perception and estimation of cutting forces. At the same time, it is difficult to obtain the exact values of the spring coefficient and damping coefficient. Therefore, it is still difficult to estimate cutting forces accurately by the force balance equation.

According to the above analysis, the corresponding relationship between the cutting force and machining signal, and the control signal of the FTS device can be obtained. To achieve an accurate estimation of cutting force in the machining process, this paper collects the machining signal, control signal, and cutting force in the machining process to train the neural network model, and thus realize an accurate estimation of cutting force.

To obtain the relationship between the extracted seven signal features and the actual cutting force (i.e., sample label), it is necessary to train the neural network to reveal the relationship. This paper uses a fully connected neural network as a machine learning model. The extracted seven features are input into the neural network through seven neurons in the input layer. After mapping between multiple hidden layers, the estimated cutting force value is output through a single neuron in the output layer. The structure of the fully connected neural network for cutting force estimation is shown in Table 1.

**Table 1.** Structure of the upper-level machine learning module for cutting force estimation.

| Layer | Neurons | Activation Function |
|---|---|---|
| Input Layer | 7 | / |
| Hidden Layer 1 | 32 | ReLU |
| Hidden Layer 2 | 16 | ReLU |
| Hidden Layer 3 | 8 | ReLU |
| Hidden Layer 4 | 4 | ReLU |
| Output Layer | 1 | / |

*4.2. Time Series Imaging of Cutting Forces*

Since the convolutional neural network is sensitive to the spatial characteristics of the signal, time series imaging is carried out on the collected time series to convert the one-dimensional signal into a two-dimensional image, so that the result can be completed by time series imaging. The method adopted in this paper is the Gramian angular summation fields (GASF) method proposed by Wang et al. [26], which can encode time series. In addition, the original time series obtained by this method can be reconstructed to understand how features are classified during the coding process. The image data are input into the HDL model.

The GASF method consists of two steps. First, the time series is represented in a polar coordinate system rather than in typical Cartesian coordinates. In this step, given $n$

observed values of a time series $x = x_1, x_2, \ldots, x_n$, and then normalized $X$, all values fall on the interval $[-1, 1]$:

$$\tilde{x}^i = \frac{x_i - \max(x) + (x_i - \min(x))}{\max(x) - \min(x)}. \tag{6}$$

The time series can be represented in polar coordinates by encoding the value as the angle of the corresponding cosine value, while the time is encoded as the radius in polar coordinates:

$$\varphi = \arccos(\tilde{x}_i), -1 \leq \tilde{x}_i \leq 1, \ \tilde{x}_i \in \tilde{x}, \tag{7}$$

$$s.t. : \ r = \frac{t_i}{N}, \ t_i \in \mathbb{N}, \tag{8}$$

where $\mathbb{N}$ is a constant factor of the span of the regularized polar coordinate system. As time goes on, the corresponding values on the polar coordinate system move between different corners. This movement preserves the time relationship and makes it easy to identify time dependencies over different time intervals. This time correlation is expressed as follows:

$$G = \begin{bmatrix} \cos(\varphi_1 + \varphi_1) & \cdots & \cos(\varphi_1 + \varphi_n) \\ \cos(\varphi_2 + \varphi_1) & & \cos(\varphi_2 + \varphi_n) \\ \vdots & \ddots & \vdots \\ \cos(\varphi_n + \varphi_1) & \cdots & \cos(\varphi_n + \varphi_n) \end{bmatrix}. \tag{9}$$

GASF images provide a time-dependent method of preservation. $G(i, j || i - j| = k)$ refers to the time interval $k$ superposition in the direction of the relative correlation. When the length of the time series is $n$, the dimension of the GASF image obtained is $n \times n$. To reduce the size of the image, the piecewise aggregation approximation (PAA) technique can be applied to smooth the time series and reduce the number of sampling points. According to the sampling points of each group of experimental data, they were converted into 512 sampling points by the PAA method and then converted into two-dimensional images by the GASF method, each of whose size was 512 × 512. After output in the form of pictures, the image can be obtained as shown in Figure 3.

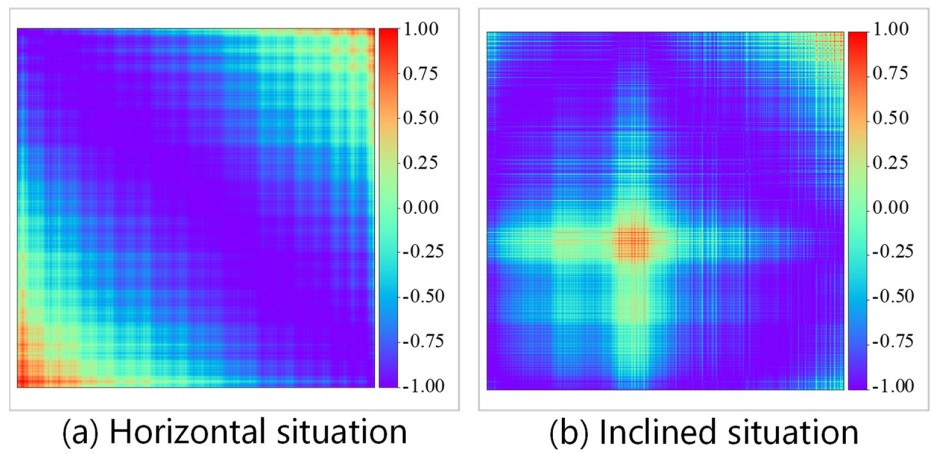

(a) Horizontal situation        (b) Inclined situation

**Figure 3.** Cutting force GASF imaging of non-wear diamond tools.

Figure 3 shows the typical examples of the GASF images of the cutting force signals of non-wear diamond tools. The images reflect the signal distribution of the cutting force within 0.6 s sampling time. The images are on the diagonal symmetry and the distribution of cutting force can be easily found. With the powerful computing power of CNN, more hidden features can be excavated.

### 4.3. The Lower-Level Customized CNN Module for Diamond Tool Wear Prediction

Based on the principle of convolutional neural network and GASF images obtained by time series imaging, an HDL is proposed for diamond-tool wear prediction. At the same time, to classify the tool wear state, it is necessary to collect the machining data after tool wear. According to the morphology quality obtained by processing, the DTW can be divided into three states, namely, no wear, slight wear, and severe wear, as shown in Figure 4. The structural parameters of the HDL model designed in this paper are shown in Table 2.

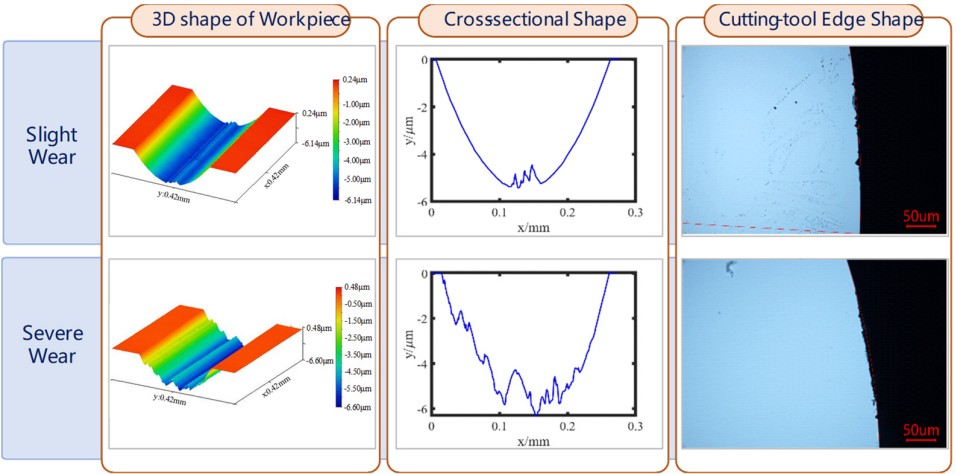

**Figure 4.** Shape of the workpiece and cutting-tool edge under different levels of wear.

**Table 2.** Parameters of lower-level customized CNN module for diamond-tool wear prediction.

| | Items | Kernel/Step Size | Input | Output | Activation Function |
|---|---|---|---|---|---|
| | Input Layer | / | $512 \times 512 \times 1$ | $512 \times 512 \times 1$ | / |
| Hidden Layer | Convolution layer 1 | $5 \times 5/1/1$ | $512 \times 512 \times 1$ | $512 \times 512 \times 64$ | ReLU |
| | Maximum pooling layer 1 | $3 \times 3/2/1$ | $512 \times 512 \times 64$ | $256 \times 256 \times 64$ | / |
| | Convolution layer 2 | $5 \times 5/1/1$ | $256 \times 256 \times 64$ | $128 \times 128 \times 64$ | ReLU |
| | Maximum pooling layer 2 | $3 \times 3/2/1$ | $128 \times 128 \times 64$ | $128 \times 128 \times 64$ | / |
| | Fully connected layer 1 | / | 1,048,576 | 384 | ReLU |
| | Fully connected layer 2 | / | 384 | 192 | ReLU |
| | Output Layer | / | 192 | 3 | softmax |

The function of the convolutional layer is to extract high-dimensional features from the input data. The convolution kernel will carry out a cross-correlation operation between the input data and the weight coefficient in the convolution kernel, and then superimpose the corresponding bias quantity, to realize the convolution operation in this region, as follows:

$$Z^{l+1}(i,j) = \left[ Z^l \otimes w^{l+1} \right](i,j) + \boldsymbol{b} = \sum_{k=1}^{K_l} \sum_{x=1}^{f} \sum_{y=1}^{f} \left[ Z_k^l(s_0 i + x, s_0 j + y) w_k^{l+1}(x,y) \right] + \boldsymbol{b},$$
$$s.t.: (i,j) \in \{0, 1, \cdots, L_{l+1}\}, \tag{10}$$

where $\boldsymbol{b}$ represents the bias of the convolution kernel; $Z^l$ and $Z^{l+1}$ represent the input and output, respectively, of the convolution operation at the $l+1$ level; $L_{l+1}$ represents the size of the output data of $Z^{l+1}$; $Z(i,j)$ represents the value of the pixel point corresponding to row $i$ and column $j$ on the pixel image; $K$ represents the number of channels in the picture; and $f$ is the size of the convolution kernel.

The size of the convolutional kernel, the size of the convolutional step, and the number of filling layers are the parameters of the convolutional layer, which together affect the output size of the convolutional layer. The influence of the hyperparameter on the output size is formulated as follows:

$$L_{l+1} = \frac{L_l + 2p - f}{s_0} + 1,$$ (11)

where $L_{l+1}$ represents the size of the output data of $\mathbf{Z}^{l+1}$; and $f$, $s_0$, and $p$ represent the size of the convolution kernel, the convolution step, and the number of filling layers, respectively. Meanwhile, activation functions (sigmoid function, tanh function, and ReLU function) need to be added after the hidden layer, whose expression is as follows:

$$A_{i,j,k}^l = f\left(\mathbf{Z}_{i,j,k}^l\right).$$ (12)

The function of the pooling layer is to reduce the dimension of the output data of the convolutional layer and carry out feature selection and information filtering. *Lp* pooling adopted in this paper is expressed as follows:

$$A_k^l(i,j) = \left[\sum_{x=1}^{f}\sum_{y=1}^{f} A_k^l(s_0 i + x, s_0 j + y)^p\right]^{\frac{1}{p}},$$ (13)

where $s_0$, $(i,j)$ have the same meaning as that of the convolution layer, and $p$ is a prespecified parameter. When $p = 1$, it is called average pooling. When $p$ goes to infinity, it is called max pooling. Meanwhile, the output data size of the pooling layer is similar to that of the convolutional layer, both of which are expressed by Equation (11).

The fully connected layer is the last part of the hidden layer of the convolutional neural network. In the fully connected layer, the two-dimensional image data will be flattened into a one-dimensional vector, thus losing the spatial topology, and then connected to other fully connected layers to transmit signals.

In the output layer, for image classification problems, the output layer generally uses normalized exponential function (softmax function) to output classification labels. In this paper, the expression of the softmax function is formulated as follows:

$$\hat{\mathbf{y}} = \text{softmax}(\mathbf{o}) \quad \hat{y}_j = \frac{\exp(o_j)}{\sum_k \exp(o_k)}$$ (14)

where $\mathbf{o}$ represents the predicted output value of the node in the output layer; $j$ indicates the number of output nodes; and $k$ represents the number of output nodes. Meanwhile, the number of output nodes represents the number of output categories. $\hat{\mathbf{y}}$ expressed after the softmax output prediction probability value belongs to the probability value of each category, and its scope is [0, 1]. Softmax does not change the order of unnormalized predicted values, but only determines the probabilities assigned to each category. Meanwhile, according to maximum likelihood estimation, the cross-entropy loss function of the classification problem can be written as follows:

$$l(\mathbf{y}, \hat{\mathbf{y}}) = -\sum_{j=1}^{q} y_j \log \hat{y}_j.$$ (15)

Among them, $\mathbf{y}$ and $\hat{\mathbf{y}}$ refer to the actual tag values and forecasting values, respectively. After the above Equation (15) is substituted, it can be obtained as follows:

$$l(\mathbf{y}, \hat{\mathbf{y}}) = \log \sum_{k=1}^{q} \exp(o_k) - \sum_{j=1}^{q} y_j o_j.$$ (16)

Using the loss function shown in the above Equations (4)–(12), its partial derivative concerning $o_j$ can be obtained as follows:

$$\partial_{o_j} l(\boldsymbol{y}, \hat{\boldsymbol{y}}) = \frac{\exp(o_j)}{\sum_{k=1}^{q} \exp(o_k)} - y_j = \text{softmax}(o)_j - y_j. \tag{17}$$

The gradient of the loss function can be obtained by applying the above Equation (17), which can be used for the training and parameter updating of the neural network.

The HDL algorithm is divided into two parts, namely, excitation propagation and weight updating. Excitation propagation includes two steps, namely, forward propagation and back propagation. Firstly, the input data are fed into the network structure through forward propagation, and then the predicted output value is obtained by passing it to the output layer. Then the partial derivative of the loss function concerning the weight of each neuron is obtained through the back propagation to form its gradient concerning the weight vector.

In the weight updating step, the weight and bias coefficient of neurons are updated according to the calculated gradient. Since the gradient represents the direction of error increase, the gradient size needs to be subtracted during updating to reduce the error, which could be formulated as follows:

$$\left(\frac{\partial \boldsymbol{E}}{\partial \boldsymbol{A}}\right)_{i,j}^{l} = \sum_{k=1}^{K_l} \sum_{x=1}^{f} \sum_{y=1}^{f} \left[ \boldsymbol{w}_k^{l+1}(x,y) \left(\frac{\partial \boldsymbol{E}}{\partial \boldsymbol{A}}\right)_{s_0 i+x, s_0 j+y, k}^{l+1} \right] f'\left(\boldsymbol{A}_{i,j}^{l}\right), \tag{18}$$

$$\boldsymbol{w}^{l} = \boldsymbol{w}^{l+1} - \alpha \left(\frac{\partial \boldsymbol{E}}{\partial \boldsymbol{w}}\right)_k = \boldsymbol{w}^{l+1} - \alpha \left[\boldsymbol{A}^{l+1} \left(\frac{\partial \boldsymbol{E}}{\partial \boldsymbol{A}}\right)_k^{l+1}\right], \tag{19}$$

where $\boldsymbol{E}$ is the error calculated by the error function; $f'$ is the derivative of the activation function, namely, the ReLU function and softmax function; $\alpha$ stands for learning rate; $\boldsymbol{A}$ represents the predicted output value; and $l$ represents the label of the hidden layer.

The above formula can be used to update the parameters of neurons and realize the training of the HDL model.

## 5. Experiments and Discussions

### 5.1. Establishing the Dataset

#### 5.1.1. Machining System

The ultra-precision machining system was set up as shown in Figure 5, which included a four-axis diamond cutting machine, a fast tool servo, and a Kistler dynamometer. The linear driver is selected as the driving device of the mechanism in the fast tool servo device. The National Instruments, PCLe-6563 acquisition card is used to realize the acquisition of the displacement signal of the grating ruler and the output of the control voltage. Thus, precise positioning and closed-loop feedback control of the device are realized. The four-axis diamond cutting machine mainly moved on the *X*-axis and *Y*-axis directions, controlling the length and speed of the cutting process. The fast tool servo controlled the movement of the diamond tool on the *Z*-axis to realize the processing of various micro/nano-morphologies. A Kistler dynamometer was used to collect cutting force signals in the machining process. The model of the dynamometer is 9109AA. The arc-shaped diamond tool was used in the experiment, and its geometrical parameters are shown in Table 3. Figure 5 shows the blade geometry of the unworn diamond tool under the optical microscope.

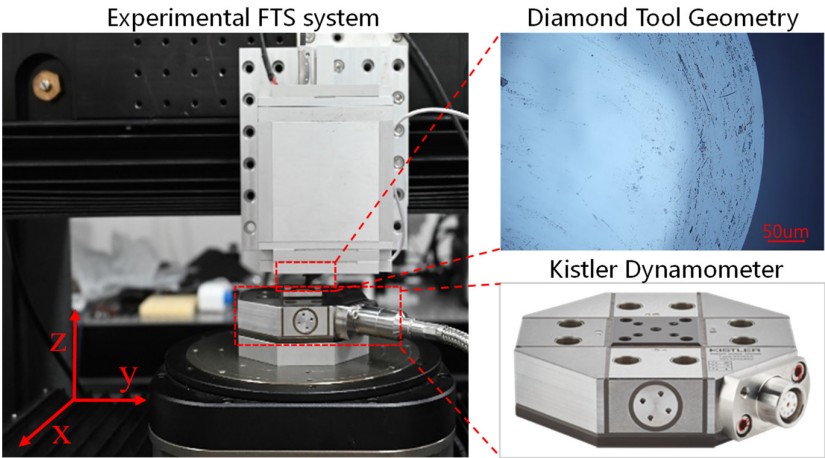

**Figure 5.** The experimental FTS system.

**Table 3.** Geometrical parameters of diamond tool.

| Radius/mm | Height/mm | Tool Angle | Rake | Clearance | Waviness |
|-----------|-----------|------------|------|-----------|----------|
| 1.028 | 4.05 | 60° | 0° | 10° | 40 nm/100° |

### 5.1.2. Experiments

Experiments were set up to collect and build a dataset for the training model. Experiments were carried out on a polished copper surface by driving the fast tool servo to cut the shape of the fixed-depth grooves on the copper surface. The processing depth of the groove cut was 5 μm, the length was 3 mm, and the processing time was 3 s. Considering that the surface of the copper sheets was inclined, the toolset was carried out on both ends of the processed groove. The slope of the copper sheet was calculated by the subtraction of different depths obtained from the toolset and the compensation for the slope of the copper sheet was added to the motion path of the tool to achieve a constant cutting depth. Figure 6 shows the morphology of the copper surface obtained under the optical microscope. To observe the depth and integrity of the morphology, a white-light interferometric microscope (NewView™ 8200, ZYGO, Middlefield, CT, USA) was used to measure the morphology, and the result is shown in Figure 6.

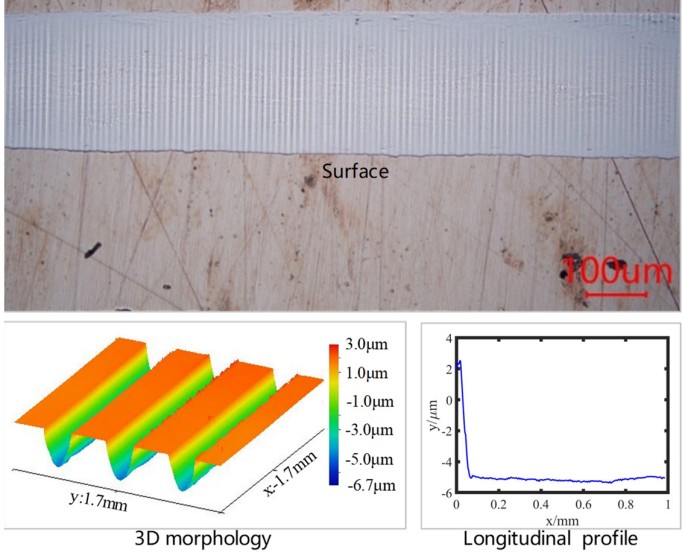

**Figure 6.** Morphology under a white-light interferometric microscope.

The Kistler dynamometer and the grating ruler inside the fast tool servo were used to collecting the machining signals in the experiment, and the sampling frequency was all 10 kHz. The datasets required for training were established after correspondence between the morphologies obtained by cutting and the experimental signals.

### 5.1.3. Data Collecting

To train a fully connected neural network model for estimating cutting forces, datasets containing machining signals, control signals, and cutting force signals were needed. To estimate the cutting forces under different cutting depths and driven frequencies, the fast tool servo was driven with different frequencies and amplitudes of the trajectories. Virtual voltages with different frequencies and amplitudes were added to the cutting process to simulate cutting forces with different amplitudes. The machining signals and control signals of the device were collected. Furthermore, the datasets for the training of fully connected neural networks were formed through the combination of virtual voltage and collected signals. The frequency and amplitude of tool position and the amplitude of virtual voltage used in the experiment are shown in Table 4. Orthogonal operations were carried out according to the data shown in Table 4. Each operation lasted 0.5 s.

**Table 4.** Frequency and amplitude of tool displacement and amplitude of the virtual voltage.

| Tool Position Frequency | Amplitude of Tool Position/μm | | | Virtual Voltage Amplitude/V | | |
|---|---|---|---|---|---|---|
| 0–90 Hz, 10 Hz/sweep | 10 | 9.9 | 9.8 | 0 | 0.5 | −0.5 |

After collecting the signal data in the experiment, it is necessary to preprocess the data to obtain the characteristic data that can be input into the neural network. The commanded velocity and command acceleration of the tool were obtained by differentiating and secondarily differentiating the command position that controlled the displacement of the diamond tool in the cutting. The actual position collected by the grating scale was differentiated and secondarily differentiated to obtain the actual velocity and acceleration. Because of the interference of high-frequency noise in the actual position and driving voltage signal, the filter of the signal was necessary. The method of directly truncating the high-frequency part of the signal was adopted to avoid the reduction of the amplitude at the specific frequency band caused by low-pass filtering. Fourier transform was used to convert the signal into a signal in the frequency domain, and then all the high-frequency parts of the signal after the specific frequency (190 Hz in the paper) were set to zero. Finally, the truncated signal was restored by the inverse Fourier transform, to avoid the problem of signal attenuation in the low-frequency part of the signal. Similarly, the actual velocity signal and actual acceleration signal obtained by differentiation were also filtered. The signals after differentiation and filtering included seven features, namely, actual position, actual velocity, actual acceleration, command position, command velocity, command acceleration, and driving voltage. Furthermore, these signal features were constructed into the datasets for the training of the neural network model. Part of constructed datasets is shown in Figure 7.

### 5.1.4. Labeling

Yan et al. [27] discussed the influence of the material stagnation region on the cutting process. The phenomenon of rising cutting forces caused by a material stagnation region was inevitable, but the cutting forces changed greatly in this case, which has a great influence on the identification of the DTW state. To avoid the influence of this phenomenon on state identification, this paper divided the variation tendency of cutting force into two cases, namely, horizontal and inclined, and then screened them. The datasets were established under different cases of cutting force, and then different neural net models were trained to identify the DTW state under the two cases, respectively.

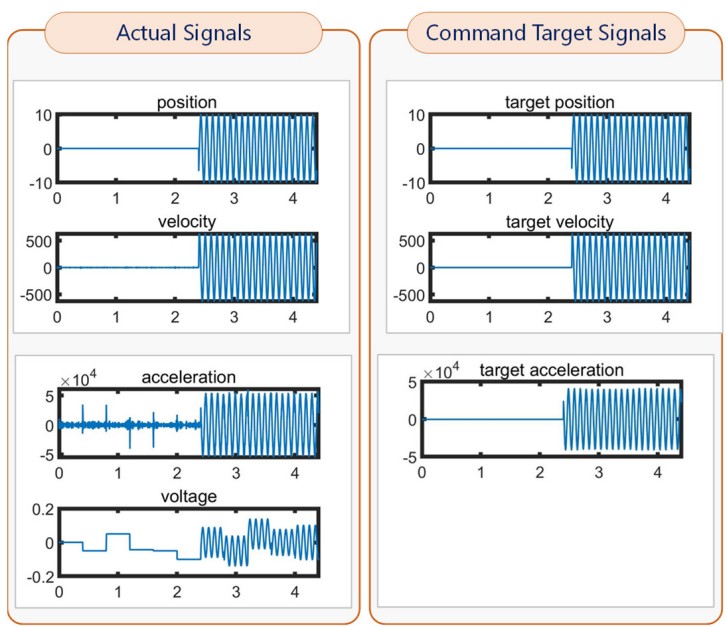

**Figure 7.** Diagrams of the input signal features.

To establish the data sets for model training, each set of selected data contained the cutting data with a processing duration of 0.2 s (i.e., there were 2000 sampling points). Furthermore, the corresponding tool wear state was taken as the label of the datasets. The established data (including three sets) are shown in Figure 8. Figure 8a,b show the estimated cutting force with the tool condition of no wear status and slight wear status. The magnitude of the force is basically same, but the force has a different tendency. At the same time, many non-time-domain signals do not show the difference visually. The estimated cutting forces were calculated by the cutting force estimation model. The estimated forces were converted into 512 × 512 two-dimensional images by the GASF imaging method and the images were integrated into tensors. The datasets contained 2238 sets of data, and the distribution is shown in Table 5.

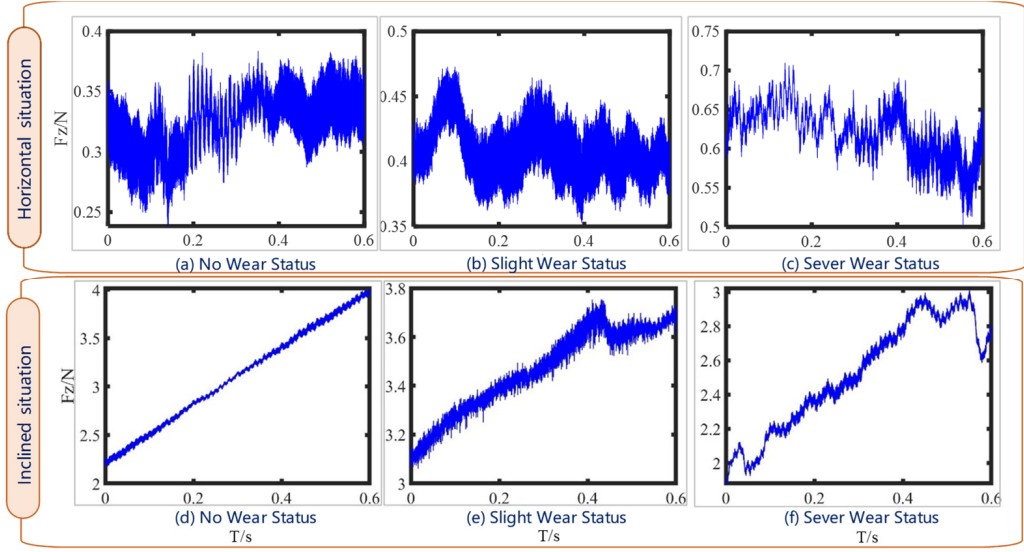

**Figure 8.** Data of different wear statuses and variation tendencies.

**Table 5.** Distribution of data samples.

|  | No Wear Status | Slight Wear Status | Severe Wear Status |
|---|---|---|---|
| Horizontal situation | 486 | 474 | 558 |
| Inclined situation | 282 | 177 | 261 |

### 5.2. Cutting Force Prediction

The preprocessed datasets were randomly divided to train the HDL model. A total of 70% of the datasets were used for the training model, and 30% were used for the verification of the model after training. In the training process, mean square error (MSE) was used as the training loss function, and the Adam algorithm was used as the optimizer of model training. The training learning rate was 0.01 and the training epochs were 7000.

The loss of the trained HDL model on the whole dataset was 0.00615, and the coefficient of determination between the estimated cutting force and the actual cutting force was 0.99685. The estimated data curve is basically the same as the training data curve when the tool is in low-frequency motion. When the tool is in high-frequency motion and subjected to low-frequency cutting force, the error of the estimation is large. It is speculated that the excessive motion frequency leads to the heating of the voice coil motor, which leads to the change of some electrical parameters. Considering that the tool motion frequency rarely reaches such a high frequency in the actual cutting situation, the method is still of high application value.

To apply the trained HDL model to the actual cutting situation, the command signals and the actual signals during the machining of the groove were collected. Then the cutting forces were estimated by the HDL model with collected signal data inputting. In addition, the resulting cutting force signals were compared with those measured by the Kistler dynamometer. By comparing the cutting forces estimated by the proposed HDL model with the cutting forces measured by the Kistler dynamometer, it can be found that the cutting forces are consistent and the trend is the same. The MSE between the estimated cutting forces and the measured cutting forces is $3.3147 \times 10^{-4}$. It can be seen that under different machining morphologies, the variation trend of the estimated cutting forces and the acquired cutting forces is the same.

To verify the method's ability to distinguish static force, variable magnitude load was applied to the experimental system by the methods that hung weights for a while and then removed them. Through the test, the static resolution of the neural network model to estimate cutting forces was measured. In the test, weights of 1 g, 2 g, 5 g, and 10 g were used for loading, respectively. The load was then removed after a certain period of loading. The measurement results of loading based on the above method are shown in Figure 9. To verify the repeatability of the model, the above weights were still used to apply loads to the experimental system. The duration of each loading was about 5 s, and each weight loading was repeated five times. The estimated forces in the loading process were obtained, as shown in Table 6. The maximum measurement error happens at the 5 g load, which is $-2.23$ mN. It can be concluded that the method has high repeatability and high measurement accuracy.

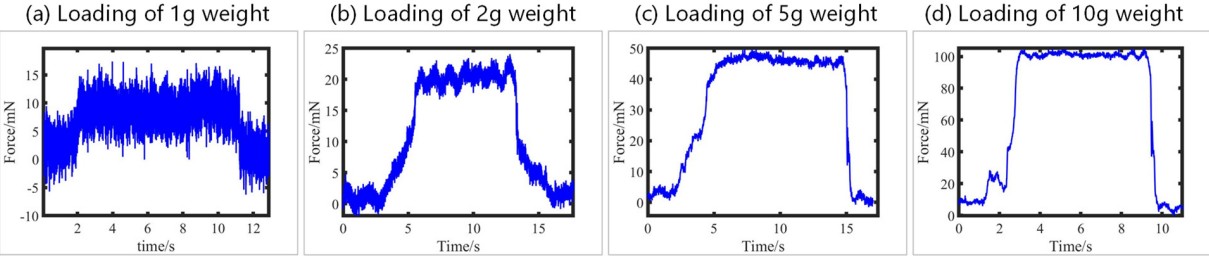

**Figure 9.** Applied load measurement results.

**Table 6.** Estimated forces under different loads.

| No. | 1 g/mN | 2 g/mN | 5 g/mN | 10 g/mN |
|-----|--------|--------|--------|---------|
| 1 | 9.4229 | 20.1711 | 48.1308 | 101.2601 |
| 2 | 10.0144 | 23.2481 | 46.5884 | 95.7065 |
| 3 | 9.4357 | 21.7028 | 46.4835 | 101.0842 |
| 4 | 8.4655 | 22.7166 | 46.1990 | 96.8934 |
| 5 | 9.4652 | 19.4296 | 46.3317 | 101.0324 |
| Avg. | 9.3607 | 21.4587 | 46.7467 | 99.1953 |

For measuring the bandwidth and the dynamic frequency response range of cutting force estimated by the model, additional experiments have been carried out. A spring was added at the bottom of the tool, and the tool was subjected to an elastic force proportional to its displacement of sinusoidal motion. The elastic force emulated the sinusoidal cutting force of different frequencies suffered by the tool. Then the cutting tool was driven to carry out sinusoidal motion of different frequencies by applying a sinusoidal sweep signal. The parameters set in the frequency sweep experiment are shown in Table 7. At the same time, the dynamometer was used to measure the elastic force at the bottom of the spring and compare it with the estimated cutting force. Part of the comparison results is shown in Figure 10.

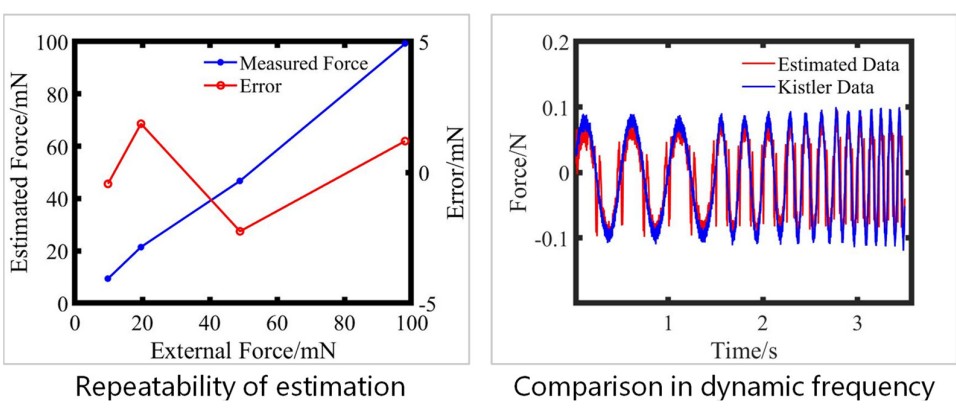

Repeatability of estimation  Comparison in dynamic frequency

**Figure 10.** Comparison in dynamic frequency and its repeatability evaluation.

**Table 7.** Experimental parameters of frequency sweeping.

| Spring Elastic Coefficient | Sinusoidal Motion Frequency | Sinusoidal Motion Amplitude | Sampling Frequency |
|----------------------------|------------------------------|------------------------------|---------------------|
| 0.02 N/μm | 1 Hz–140 Hz, 1 Hz/sweeping | 5 μm | 10 kHz |

Through statistical analysis of different cutting forces obtained by the model estimation, the dynamic frequency response diagram of force estimation as shown in Figure 11 can be obtained. The spring elastic coefficient obtained by the estimation method is the same as the actual elastic coefficient when the frequency range is lower than 70 Hz, while there is a maximum value at 70 Hz, which is the resonance caused by the drive frequency reaching the natural frequency of the device itself. When the frequency range is higher than 100 Hz, the elastic coefficient of the high-frequency part rises again due to the poor estimation ability of the estimation method for the high-frequency signal. It can be concluded that the bandwidth of the estimated force is about 70 Hz, and its bandwidth is mainly limited by the natural frequency of the system itself.

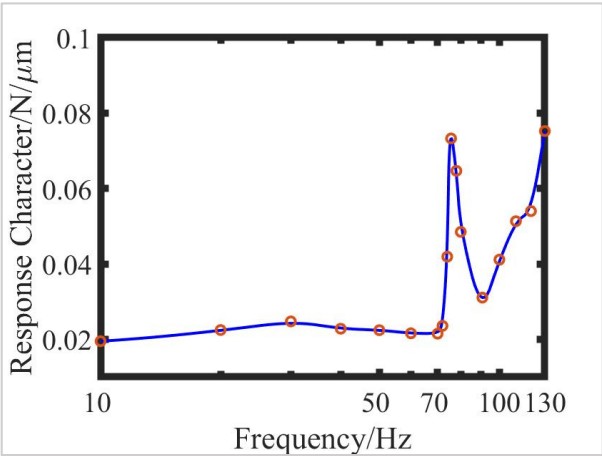

**Figure 11.** Dynamic frequency response diagram of cutting force estimation.

Comparing Figure 12 with Figure 11, it can be found that the cutting forces in Figure 12 differ greatly at the trough of the sinusoidal force. The MSE of the estimated cutting forces and the cutting forces collected by the Kistler dynamometer are 0.0027 and 0.0110, respectively. Thus, the cutting force estimation method based on HDL is more accurate. The reason is that the position of the tool changes all the time when machining sinusoidal morphology. The driving force of the voice coil motor is different from the linear model because of its nonlinearity, which leads to a larger error in the estimation of cutting force. The HDL model is more accurate when facing the nonlinear driving force of the voice coil motor because of its ability to deal with the nonlinear system.

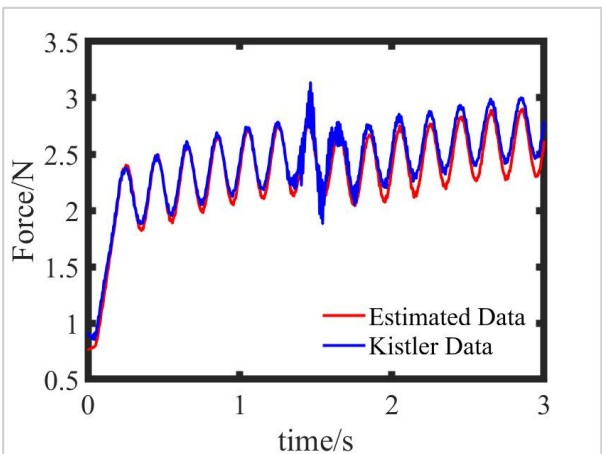

**Figure 12.** Cutting force of sinusoidal morphology estimated by system identification method.

### 5.3. Cutting-Tool Wear Classification

For the implementation of the HDL algorithm, the hardware used in this paper comprises six Intel Xeon Gold 6142 processors (3.5 GHz, 27.1 GB, Intel Corporation, Santa Clara, CA, USA) and NVIDIA RTX 3080 graphics processors (10.5 GB, NVIDIA Corporation, Santa Clara, CA, USA). The Stochastic Gradient Descent (SGD) optimizer is used to train the HDL model, and the fixed step-size decay (StepLR) learning strategy is used to make the model converge quickly in the later stage. The parameter settings for HDL model training are shown in Table 8.

**Table 8.** Parameter settings of HDL.

| Parameter | Learning Rate | Weight Decay | Step Size | Epoch Times | Batch Size |
|---|---|---|---|---|---|
| Value/Type | 0.01 | 0.001 | 10 | 100 | 5 |

The data results obtained through training are shown in Table 9. The classification accuracies of the models trained from the force data collected by the Kistler dynamometer were all higher than the models trained from the estimated force data. The resolution on the cutting force of the dynamometer is better than the HDL-based cutting force estimation model, which may lead to the acquisition of force signals being more accurate than the estimated force signals. At the same time, there are also many noises in the machining signals obtained by the grating ruler and acquisition card. These noises may also have a great influence on the classification results, which results in low classification accuracy. Through the training results, it can be seen that the errors of the trained HDL model on the training set are close to 0, and the classification accuracy on the training set and the test set is higher than 85%. However, the accuracy of the two models on the training set is higher than that on the test set, indicating that the models still have the phenomenon of overfitting, and the generalization ability of the models still needs to be improved. The classification errors mainly appeared in the classification of no wear and slight wear, but the classification errors for severe wear were few. The reason is that the amount of datasets used for training is small. To obtain a model with stronger generalization ability, the datasets need to be further augmented.

**Table 9.** Comparison of model classification results on cutting force.

| Accuracy | HDL-Based Estimated Results | | Results Collected by Dynamometer | |
|---|---|---|---|---|
| | Horizontal Case | Inclined Case | Horizontal Case | Inclined Case |
| Training | 100.0% | 99.5% | 100.% | 100.0% |
| Test | 89.3% | 86.6% | 97.4% | 94.0% |

The HDL proposed in this paper was compared with other neural network models including AlexNet and VGG. The estimated cutting force data were input into these other networks. The classification results of the three networks are shown in Table 10.

**Table 10.** Classification results of different models.

| | HDL | | AlexNet | | VGG | |
|---|---|---|---|---|---|---|
| Cases | Horizontal | Inclined | Horizontal | Inclined | Horizontal | Inclined |
| Accuracy | 89.3% | 86.6% | 84.3% | 79.4% | 75.7% | 80.1% |

Through the comparison of classification results, it can be seen that the HDL model proposed in this paper has higher accuracy classification results than other network models. AlexNet has fewer network structure parameters (occupying 1.4 GB memory) than the network structure parameters used in this paper (occupying 1.5 GB memory), and its convergence speed in the training process is slower. While the network structure parameters of VGG (occupying 1.6 GB memory) are more complex, the main reason for its low classification accuracy is the phenomenon of overfitting, indicating that the network structure is too complex compared to the amount of training datasets. Therefore, the HDL model proposed in this paper is more consistent with the amount of datasets collected, so the training effect is better.

*5.4. Discussions*

According to the experimental results, it can be seen that the trained HDL model can accurately distinguish the state of DTW. However, limited by the amount of training datasets, the accuracy of the model still needs to be improved. At the same time, due to the limited wear tool samples used in the collected datasets, the model's identification ability of slight diamond-tool wear is poor. Due to the lack of samples with smaller wear amounts and the limitation of the generalization ability of the HDL model, the trained model cannot judge the specific state of the diamond tool whose wear amount is tiny. The minimum tool wear amount that can be identified by the model can only reach the front-tool-face wear amount of about 1.5 μm. The classification accuracy will decrease greatly when the HDL model deals with a diamond tool with a wear less than 1.5 μm on the front tool face. It is still necessary to further expand the amount of the dataset and the number of diamond-tool samples to train a model that can distinguish the diamond-tool states with smaller wear. On the other hand, future research could be conducted on introducing deep reinforcement learning techniques [28,29] to the DTW and its following-up decisions.

Due to different wear mechanisms when cutting different materials, the cutting force signals are greatly different. Thus, it is difficult for same HDL model to classify tool wear condition accurately with different materials. To make the model adapt for different materials, a meta-learning module, which can fine-tune the parameters of the HDL model with different cutting conditions, can be added in this HDL model.

## 6. Conclusions

This paper presented a hybrid deep learning (HDL) model as the core of a digital twin system to predict the wear state of diamond tools in the ultra-precision machining process. A self-sensing cutting force estimation method is established based on machine learning. According to the machining signal and control signal of the FTS device in the ultra-precision machining process, the cutting force signal is estimated by using a multi-layer fully connected neural network trained by the special machining process signal. This method does not need to rely on a force sensor and can achieve an accurate estimation of cutting force. Combined with the cutting force estimation method, the motion displacement, velocity, and other machining signals of the FTS device are used to estimate and reconstruct cutting force. Since the convolutional neural network is sensitive to the spatial characteristics of the signal, a one-dimensional signal of estimated cutting force is converted into a two-dimensional signal by the time series imaging method. The transformed two-dimensional image data are input into the HDL model for predicting the wear state of the diamond tool. Future research directions include the refining of data modeling and the introduction of advanced deep reinforcement learning techniques.

**Author Contributions:** Conceptualization, B.J. and Y.C.; methodology, L.W. and Y.C.; software, Y.T., K.S. and L.W.; validation, L.W., Y.T., K.S. and Y.C.; formal analysis, L.W.; investigation, Y.T., L.W. and K.S.; resources, Y.C.; data curation, Y.T., K.S. and L.W.; writing—original draft preparation, L.W. and K.S.; writing—review and editing, L.W.; visualization, L.W.; supervision, Y.C. and B.J.; project administration, Y.C.; funding acquisition, Y.C. All authors have read and agreed to the published version of the manuscript.

**Funding:** This research was funded by the National Key R&D Program of China under Grant 2020YFB2007600, in part by the Ministry of Industry and Information Technology's Manufacturing High-quality Development Project under Grant TC200H02J, and in part by the National Natural Science Foundation of China under Grant 51975522. B.F. Ju appreciates the support from the National Natural Science Foundation of China (Grant Nos. 52035013 and U1709206), the Science Fund for Creative Research Groups of National Natural Science Foundation of China (No. 51821093), and the Zhejiang Provincial Key R&D Program of China (No. 2018C01065).

**Institutional Review Board Statement:** Not applicable.

**Informed Consent Statement:** Not applicable.

**Data Availability Statement:** Data are unavailable due to privacy or ethical restrictions.

**Conflicts of Interest:** The authors declare no conflict of interest.

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
