# Peer review of "A Hybrid Deep Learning Model as the Digital Twin of Ultra-Precision Diamond Cutting for In-Process Prediction of Cutting-Tool Wear"

_applsci, doi:10.3390/app13116675_

Round 1

Reviewer 1 Report

Manuscript Number: applsci-2364056

Title: A hybrid deep learning model as the digital twin of ultra-preci-sion diamond cutting for in-process prediction of cutting-tool wear

Decision: Minor revision

Article Type: Article

The article is, in general, well written but there are some issues that article should consider to revise in order to improve its quality. Some comments were done in this way:

Ø  The abstract, according to the reviewer, is not a mini-paper but a quick tool to help readers decide whether they will read the rest of the paper. Please give numerical (percentage) improvements to the summary section that will attract the attention of the readers.

Ø  Figure 6 Color scale should be given for 3D morphology.

After making the above corrections would recommend this article for publication in Applied Sciences.

Reviewer 2 Report

In-process measurement of tool wear in ultraprecision machining is attracting much attention. This paper proposed a hybrid deep learning model based on cutting force for tool wear state prediction in ultra-precision diamond cutting. This topic is interesting, and the results have shown that the model can monitor the tool wear well. Some comments are as follow:

1.   In page 3, line 117, authors developed a fast tool servo device driven by voice coil motor, but in Fig. 1 “control principle of FTS”, piezoelectric ceramics was mentioned. What these piezoelectric ceramics used for?

2.   In page 4, line 156, there is a typing error in “Lorentz force (Fdring)”.

3.   What information can we get from Fig. 3? Text should be added to explain this figure.

4.   Is FTS device-based force algorithm only able to predict one directional force (thrust force)? A comparison of the force generated by using worn and unworn tools under the same machining conditions may help reader understand why this force is able to estimate tool wear.

In other words, the characteristics of the forces used to determine the appearance of wear on a tool should be given.

5.   How many data are needed to train HDL model for predicting tool wear? What these data look like? If possible, list several typical data that was used to train.

6.   What is the smallest tool wear that can be identified by this HDL model?

7.   Does the HDL model need to be retrained when estimating tool wear in cutting different materials?

8.   In this study, copper was used as workpiece. Can this model be used in the cutting of brittle materials? What kinds of modifications need to be made. The authors can give an outlook on this.

The English language is good.

Reviewer 3 Report

The source of figure 1 must be indicated if it is not from the authors of the paper.

Author Response

Thank you for your kind comment. We apologize for the controversial figure. Figure 1 has been modified in the revised version of the manuscript and the source has been added in the figure.